# Vector Competence of the Invasive Mosquito Species *Aedes koreicus* for Arboviruses and Interference with a Novel Insect Specific Virus

**DOI:** 10.3390/v13122507

**Published:** 2021-12-14

**Authors:** Stephanie Jansen, Dániel Cadar, Renke Lühken, Wolf Peter Pfitzner, Hanna Jöst, Sandra Oerther, Michelle Helms, Branka Zibrat, Konstantin Kliemke, Norbert Becker, Olli Vapalahti, Giada Rossini, Anna Heitmann

**Affiliations:** 1Department of Arbovirology, Bernhard Nocht Institute for Tropical Medicine, 20359 Hamburg, Germany; jansen@bni-hamburg.de (S.J.); danielcadar@gmail.com (D.C.); luehken@bnitm.de (R.L.); joest@bnitm.de (H.J.); helms@bnitm.de (M.H.); zibrat@bnitm.de (B.Z.); Konstantin.kliemke@bnitm.de (K.K.); 2Faculty of Mathematics, Informatics and Natural Sciences, University of Hamburg, 20148 Hamburg, Germany; 3Kommunale Aktionsgemeinschaft zur Bekämpfung der Schnakenplage e. V. (KABS), 67346 Speyer, Germany; wolf-peter.pfitzner@kabs-gfs.de; 4Institute for Dipterology, 67346 Speyer, Germany; sandra.oerther@gmail.com (S.O.); norbertbecker@gmx.de (N.B.); 5Institute of Global Health, Medical Faculty Heidelberg University, 69117 Heidelberg, Germany; 6Centre for Organismal Studies (COS), Faculty of Life Sciences, Heidelberg University, 69711 Heidelberg, Germany; 7Icybac Mosquitocontrol GmbH, 67346 Speyer, Germany; 8Department of Virology, Faculty of Medicine, University of Helsinki, 00014 Helsinki, Finland; olli.vapalahti@helsinki.fi; 9Unit of Microbiology, Department of Experimental, Diagnostic and Specialty Medicine, University of Bologna, 40138 Bologna, Italy; giada.rossini@unibo.it

**Keywords:** invasive mosquito species, *Aedes koreicus*, arbovirus transmission, vector competence, insect specific virus, Wiesbaden virus

## Abstract

The global spread of invasive mosquito species increases arbovirus infections. In addition to the invasive species *Aedes albopictus* and *Aedes japonicus, Aedes koreicus* has spread within Central Europe. Extensive information on its vector competence is missing. *Ae. koreicus* from Germany were investigated for their vector competence for chikungunya virus (CHIKV), Zika virus (ZIKV) and West Nile virus (WNV). Experiments were performed under different climate conditions (27 ± 5 °C; 24 ± 5 °C) for fourteen days. *Ae. koreicus* had the potential to transmit CHIKV and ZIKV but not WNV. Transmission was exclusively observed at the higher temperature, and transmission efficiency was rather low, at 4.6% (CHIKV) or 4.7% (ZIKV). Using a whole virome analysis, a novel mosquito-associated virus, designated Wiesbaden virus (WBDV), was identified in *Ae. koreicus.* Linking the WBDV infection status of single specimens to their transmission capability for the arboviruses revealed no influence on ZIKV transmission. In contrast, a coinfection of WBDV and CHIKV likely has a boost effect on CHIKV transmission. Due to its current distribution, the risk of arbovirus transmission by *Ae. koreicus* in Europe is rather low but might gain importance, especially in regions with higher temperatures. The impact of WBDV on arbovirus transmission should be analyzed in more detail.

## 1. Introduction

The transmission of viruses by mosquitoes (arthropod-borne viruses), represent a threat for global health, with increasing numbers of infections reported in the last few decades [1]. One determinant factor represents the importation and the establishment of arboviruses outside their natural area of origin, which can lead to local or regional epidemics. Globalization and international mobility accelerate the migration of exotic pathogens to new environments, facilitating contacts with susceptible new hosts. Indigenous mosquito populations can be infected and, thereby, transmit arboviruses to immunologically naïve amplification hosts, which may cause autochthonous epizootics or epidemics. Recently, the importation of Zika virus (ZIKV) from Asia to the Americas led to an epidemic with hundreds of thousands of human cases in 2015/2016 [2]. Another example is the global spread of chikungunya virus (CHIKV) in the last two decades. It initially only caused epidemics in Africa and parts of Asia, but afterward spread to India, the Indian Ocean islands and the Americas, causing millions of human infections [3]. A single introduction of West Nile virus (WNV) to the US in 1999, commonly distributed in Africa, Europe, the Middle East, West Asia and Australia, was followed by a rapid spread over the whole North American continent, with thousands of human neuroinvasive cases and more than 2000 deaths [4].

Another important factor for the increasing number of arbovirus infections is the global spread of invasive mosquito species, particularly driven by the global trade, such as *Aedes albopictus* and *Aedes japonicus* in the last few decades [5,6]. These invasive species can act as vectors for viruses, which are not transmitted by indigenous mosquito species. For instance, all reported outbreaks of autochthonous CHIKV transmission in Europe are in areas where *Ae. albopictus* is present [7]. Recently, another invasive species was introduced to Europe: *Aedes koreicus*. This species is endemic in Japan, South Korea and Far Eastern parts of China and Russia [8]. Like *Ae. japonicus* and *Ae. albopictus*, *Ae. koreicus* colonizes in different types of natural and artificial containers. The larvae are found in urban areas, as well as in forested areas [9]. Although *Ae. koreicus* and *Ae. albopictus* can coexist in the same habitat, larval competition is weak [10]. With an average temperature of 11.5 °C in the native habitat of *Ae. koreicus* and the coldest month of −9 °C, *Ae. koreicus* is certainly able to handle the climatic conditions of Central Europe [11]. *Aedes koreicus* mosquitoes are day active. Even though *Ae. koreicus* can use a wide range of different mammalophilic hosts, *Ae. koreicus* seems to act mainly anthropophilic [12]. The first report of *Ae. koreicus* in Europe was in 2008 in Belgium, which is generally the first detection outside of the native area. The way of introduction is unknown, assumedly by international trade [13]. In the following years, the *Ae. koreicus* population was able to establish in that area without range expansion. In contrast, the first detection of an *Ae. koreicus* population in a region in northern Italy in 2011 was followed by a rapid spread in that region [14,15]. The first detection of *Ae. koreicus* in Germany was in 2015 within the context of a citizen science project; the first established population was described in 2016 in southern Germany [16,17]. Further detection of *Ae. koreicus* in Europe followed in southwest Russia, Hungary, Slovenia, Switzerland and Austria [18,19,20,21,22].

Thus, there is an urgent need to elucidate the vector competence of *Ae. koreicus* for arboviruses. In laboratory studies, *Ae. koreicus* was able to transmit the parasite *Dirofilaria immitis* [23]. Moreover, *Ae. koreicus* is assumed to transmit Japanese encephalitis virus (JEV), with JEV-positive specimens caught in the field [24]. The only laboratory study for arbovirus transmission by *Ae. koreicus* was performed with CHIKV. *Ae. koreicus* from Italy was revealed as a potential vector for CHIKV but only with a low transmission efficiency at 23 °C (5%) and no positive saliva under a fluctuating temperature of 12–27 °C [25].

Recent advances in next-generation sequencing (NGS) opened up new insights into the virome characterization of mosquitoes, which presumably has an influence on the vector competence of mosquitoes [26]. Almost all investigated mosquitoes harbor novel viruses thatbelong to families that are assumed to be insect-specific viruses (ISV), but there are also some that are reported for plants or vertebrate hosts [27]. There is little known about the influence of these novel mosquito-related viruses on the fitness of mosquitoes. Likewise, there is also little known about the impact of ISVs on the transmission of arboviruses. Some studies showed an enhanced effect, while others showed a decreasing effect on vector competence [28,29]. The decreasing effect on vector competence could be useful for arbovirus control. Thus far, nothing has been reported about the virome of *Ae. koreicus* mosquitoes.

In light of the continuing spread of *Ae. koreicus* in Europe and the major lack of knowledge about the vector capacity, we investigated the vector competence of field-caught *Ae. koreicus* mosquitoes from Germany for several arboviruses. This includes WNV, which is endemic in southeastern Europe and was detected in Germany for the first time in 2018 [30]; CHIKV, an emerging arbovirus with regular spatial-restricted outbreaks in Italy and France; and ZIKV, with the first autochthonous cases in Europe reported in 2019 in France [31]. Second, we analyzed the virome of the investigated *Ae. koreicus* mosquitoes from Germany and evaluated the influence of coinfection with a novel mosquito-associated virus on vector competence.

## 2. Materials and Methods

### 2.1. Vector Competence Studies

#### 2.1.1. Collection and Rearing of Mosquitoes

*Aedes koreicus* eggs were collected with ovitraps in Western Germany on cemeteries (50°03′ N, 8°16′ E/50°05′ N, 8°16′ E/50°08′ N, 8°17′ E) in summer 2018 and 2019 [17]. Sticks with attached dry eggs were flooded for two days, air-dried for two days and again flooded for two days. Mosquitoes were reared at 26 °C with a relative humidity of 80% and a light:dark period of 12:12 h, including twilight for 30 min. Taxonomic identification of larvae (4th larval stage) was performed as described by Pfitzner et al. [17]. Ten randomly selected adult specimens were analyzed via pan-Flavivirus, -Alphavirus and -Orthobunyavirus PCR to exclude other arbovirus infections [32]. All pan-PCRs were negative.

#### 2.1.2. Infection of Mosquitoes

Experimental infections on vector competence were performed in the BSL-3 insectary at the Bernhard Nocht Institute for Tropical Medicine, Hamburg, Germany. Vials of twenty 3–14-day-old F0 females were sorted and starved 24 h before an artificial blood meal. For reaching a high feeding rate, prewarmed blood (37 °C) was offered via two 50 µL droplets per vial for two hours, resulting in a feeding rate of 67% (i.e., the percentage of engorged females per the total number of females). Infectious blood meals contained 50% human blood (expired banked blood), 30% fructose solution (stock concentration 8%), 10% filtrated bovine serum and 10% virus stock. CHIKV blood meal (strain CNR_24/2014 European Virus Archive goes Global project, ECSA lineage, fifth passage) had a final concentration of 10^6^ plaque-forming units per mL (pfu/mL). ZIKV (Genbank KU870645.1, fifth passage) [33] and WNV lineage 1 (Genbank HM991273/HM641225, fifth passage) [34] were offered with a final concentration of 10^7^ pfu/mL. Engorged females were incubated at 24 h-fluctuating temperatures of 27 ± 5 °C or 24 ± 5 °C to mimic day–night variations, with a humidity of 70% and fed by fructose-soaked cotton pads, which were refreshed with new fructose every 2–3 days.

#### 2.1.3. Infection Analysis

Fourteen days post-infection, mosquitoes were screened for infection rate (IR), transmission rate (TR) and transmission efficiency (TE), as described previously [32]. Briefly, the IR is defined as the number of virus-positive mosquito bodies per number of fed females, TR as the number of virus-positive saliva per the number of virus-positive bodies and TE is calculated as the number of virus-positive saliva per the total number of fed and analyzed specimens. Detection and quantification of RNA was done by quantitative real-time PCR assay (RealStar Chikungunya RT-PCR Kit 2.0; RealStar Zika Virus RT-PCR Kit 1.0; RealStar WNV RT-PCR Kit 1.0; altona Diagnostics, Hamburg, Germany). A salivation assay was performed as previously described, detecting viable virus particles in the saliva by incubation of saliva solution on Vero cells (*Chlorocebus sabaeus*; CVCL_0059, obtained from ATCC, Cat# CCL-81) [35]. The R program [36] was used for all data analysis and visualizations, including the *readxl* [37], stringr [38], *dplyr* [39], *plyr* [40] and *ggplot2* [41] packages.

### 2.2. Detection and Genomic Characterization of a Novel Insect-Specific Virus

#### 2.2.1. Metagenomic and Metatranscriptomic Analysis

A pool of 20 *Ae. koreicus* specimens were subjected to metagenomic and metatranscriptomic characterization of viral components. Briefly, after filtration through a 0.45 µm filter (Millipore, Darmstadt, Germany), 200 µL of mosquito homogenate was treated with a mixture of nucleases (RNAse One, Promega, Fitchburg, WI, USA; Turbo DNase, Ambion, Carlsbad, CA, USA) to digest unprotected nucleic acids, including host DNA/mRNA and some bacteria. A MagMAX™ Viral RNA Isolation Kit (Life Technologies, Carlsbad, CA, USA) was used to extract RNA/DNA following to the manufacturer’s instructions. After reverse transcription and cDNA synthesis, the QIAseq FX DNA Library Kit (Qiagen, Hilden, Germany) was used for library preparation and subsequently subjected for sequencing on a MiSeq Illumina platform (Illumina, San Diego, CA, USA). Resulting raw reads were first qualitatively checked and filtered, followed by de novo assembly separately using Trinity. The contigs were BLASTx against viral and a nonredundant proteome database (NCBI; Bethesda, MD, USA) with an E-value cutoff of 0.001. The virus-like output of contigs and singlets were visualized and analyzed in MEGAN [42]. Genome assembly, open reading frames (ORF) and nucleotide and amino acid sequence analyses were performed using Geneious v9.1.7. (Biomatters, Auckland, New Zealand) and ORFfinder (https://www.ncbi.nlm.nih.gov/orffinder/, accesed date: 1 September 2021). The evolutionary relationship of the newly detected virus was analyzed by inferring a phylogenetic tree based on amino acid sequences of the RdRp gene using the maximum likelihood (ML) method in SeaView v4 [43], with a Blosum + I + G substitution model.

Four supernatants from Wiesbaden virus (WBDV) positive *Ae. koreicus* pools were inoculated with semi-confluent *Ae. albopictus* cells (C6/36; CVCL_Z230, Friedrich-Loeffler-Institute, Riems, Germany). Insect cells were incubated at 27 °C for one week, and no signs of cytopathic effects (CPE) could be observed. Supernatant as well as cells were passaged and subjected to virus-specific RT-PCR. Passaging was performed four times, and CT values between the passages were consistent.

#### 2.2.2. RT-PCR for Detection of Novel Insect Specific Virus

RT-PCRs were performed using a Superscript III One-Step RT-PCR Kit (Invitrogen, Carlsbad, CA, USA), with a specific primer for the novel insect-specific virus provisionally designated Wiesbaden virus (WBDV): WBDV-F: CCATGTCCCGATCAGTTGTA/WBDV-R: CGTCAACTCCTTCAACTGTG. Cycling conditions were as follows: 60 °C for 1 min, followed by 50 °C for 45 min and an initial denaturation at 94 °C for 2 min. The following 45 PCR cycles comprised of 94 °C for 15 s, followed by 55 °C for 30 s and 68 °C for 30 s, concluded by a final extension step of 68 °C for 7 min.

#### 2.2.3. Screening of Additional Mosquito Species for WBDV

We have investigated wild caught *Aedes sticticus* (2018, southern Germany, 49°26′ N, 8°24′ E/49°28′ N, 8°25′ E), *Anopheles maculipennis* s.l. (2018, southern Germany, 49°26′ N, 8°24′ E/49°28′ N, 8°25′ E), *Aedes detritus* (2018, Baltrum; 53°44′ N, 7°22′ E), *Aedes japonicus* (2018, southern Germany, 49°31′ N, 8°40′ E), *Culex torrentium* (2018, southern Germany, 49°32′ N, 8°40′ E), *Culex pipiens* biotype *pipiens* (2018, northern Germany; Lon: 53.467821/Lat: 9.831346) and lab colonies of *Culex pipiens* biotype *molestus* (established from a population in southern Germany in 2011), *Culex quinquefasciatus* (long established lab colony) and *Ae. albopictus* from Germany (established from a population in Freiburg in 2015) and from Italy (established from a population in Calabria in 2015) for the presence of WBDV. Mosquitoes were homogenized and extracted individually; RNA was analyzed via RT-PCR, as described. The analysis was repeated in 2020 with 30 specimens of *Ae. Koreicus* and 30 individuals of *Aedes geniculatus* from the same place in southwest Germany (50°03′ N, 8°16′ E/50°05′ N, 8°16′ E/50°08′ N, 8°17′ E) and 30 specimens of *Ae. Japonicus* mosquitoes, collected in southwest Germany (49°31′ N, 8°40′ E). All mosquito specimens that were investigated in the vector competence studies were screened via RT-PCR for WBDV.

#### 2.2.4. Statistical Analysis of Coinfection

For the infection experiments with the arboviruses CHIKV and ZIKV, WBDV was found more frequently, allowing us to analyze the coinfection patterns and the impact of coinfection on the CHIKV and ZIKV susceptibility of *Ae. koreicus* specimens. Due to the low TRs, we only focused on the virus infections and arbovirus body titers. A Chi-square test was used for a bivariate analysis to determine the proportion of infected specimens per arbovirus in relation to the proportion of WBDV-infected specimens. A generalized model was used to analyze the differences in the arbovirus titres for coinfected and non-coinfected specimens, with titer as a response variable and WBDV-infection status as an explanatory variable in R [36].

## 3. Results

### 3.1. Vector Competence Studies

*Aedes koreicus* specimens collected in southwest Germany were able to transmit CHIKV (Table 1). Transmission was observed at 27 ± 5 °C, with an infection rate (IR) of 68% and a transmission rate (TR) of 7%, indicating a transmission efficiency (TE) of 5%. Incubation at a lower temperature of 24 ± 5 °C resulted in an IR of 17.6%, but no transmission was observed. The analysis of ZIKV infection showed a similar picture; *Ae. koreicus* mosquitoes were able to transmit ZIKV only at a higher temperature (Table 1). At a temperature of 27 ± 5 °C, mosquitoes transmitted ZIKV with an IR of 81%, a TR of 6% and a TE of 5%. The IR at 24 ± 5 °C had a slightly lower ratio of 79%, while no transmission was observed. In addition, *Ae. koreicus* mosquitoes were susceptible for WNV infection, but no transmission was observed (Table 1). The IR value was 85% at 27 ± 5 °C and 9% at 24 ± 5 °C, respectively.

### 3.2. Identification of a Novel Insect-Specific Virus

We observed a high number of viral reads related to luteo- and sobemo-like viruses. These contigs were assembled, and a complete genome of a novel virus was successfully recovered, which we provisionally named Wiesbaden virus (WBDV), after the city where the eggs were collected. The WBDV genome consists of two segments, which include two ORFs on RNA1 (RdRp and a hypothetical protein) and a single ORF on RNA2 (capsid) (Figure 1). The ORFs on RNA1 share the highest identity with Atrato Sobemo-like virus 4 (RdRp, 86%; hypothetical protein, 61%), and the remaining capsid protein in RNA2 is most similar to Atrato Sobemo-like virus 4 (73%). The phylogenetic analysis of the RdRp protein clustered the WBDV in a common clade with other mosquito-associated luteo-/sobemo-like viruses (Figure 1). The infection of two mosquito cell lines (C6/36 and Aag2 (*Ae. aegypti*; CVCL_Z617, Friedrich-Loeffler-Institute, Riems, Germany) with a WBDV-positive mosquito homogenate was successful, and no cytopathic effect was observed. Screening of the investigated *Ae. koreicus* specimens revealed an IR for a WBDV infection between 0 to 20.9% (Table 1).

#### 3.2.1. Correlation of Arbovirus = WBDV Coinfection

From all 185 investigated mosquitoes in the CHIKV vector competence studies, 13 were positive for WBDV and 12 of these were coinfected with CHIKV. A total of 172 of the investigated specimens were negative for WBDV, of whom 97 were CHIKV-positive. At 92.3%, the proportion of CHIKV-positive specimens with WBDV coinfection (12/13) was statistically significantly higher than the proportion of WBDV-negative and CHIKV-positive specimens (97/172) at 56.1% (Chi-square test: χ2 = 5.1121, df = 1, *p* = 0.02). No statistically significant effect was found for ZIKV (100% (9/9) vs. 77.8% (56/72); Chi-square test: χ2 = 1.2875, df = 1, *p* = 0.26) (Table 1). Likewise, titres for CHIKV were statistically significant higher in WBDV-positive specimens (Deviance = 145.81, df = 1, *p* = 0.004), but no correlation was observed for ZIKV (Deviance = 23.156, df = 1, *p* = 0.12) (Figure 2).

As only one mosquito of all investigated specimens in the WNV experiments (1/73; Table 1) was WBDV positive, no correlation of WNV with WBDV infection could be calculated.

#### 3.2.2. Screening of Different Mosquito Species for Wiesbaden Virus

Ten different mosquito species were screened for WBDV in 2018. *Aedes detritus*, *Aedes sticticus*, *Anopheles maculipennis* s.l., *Culex. pipiens* biotype *pipiens* (northern Germany), *Culex torrentium* (southern Germany), *Culex pipiens* biotype *molestus*, *Ae*. *albopictus* from Germany and Italy, as well as *Culex quinquefasciatus* were negative for WBDV. The only species with positive results for WBDV was *Ae. japonicus* (7%; 2/30).

Screening for WBDV was repeated in 2020 for *Ae. koreicus* and additionally for two species from a related habitat: *Aedes geniculatus* and *Ae. japonicus*. WBDV infection could be detected in all three species, and the infection rate of 30 individuals per species was 53% for *Ae. koreicus* (16/30), 67% for *Ae. geniculatus* (20/30) and 100% for *Ae. japonicus* (30/30).

## 4. Discussion

Similar to previous studies from Ciocchetta et al., *Ae. koreicus* specimens collected in southwest Germany were able to transmit CHIKV [25]. Transmission was only observed at a higher temperature of 27 ± 5 °C, with no transmission at 24 ± 5 °C. Accordingly, the IR at 27 ± 5 °C and 68.2% was four times higher than at 24 ± 5 °C and 17.6%. This matches with Ciocchetta et al., also showing a temperature-dependent transmission of CHIKV by *Ae. koreicus* from Italy, with transmission efficiencyies also in the lower-level of around 5% (3 and 10 dpi) [25]. In contrast, other mosquito species, such as *Ae. albopictus* from Germany and Italy, are showing no temperature dependence for transmission [7].

To the best of our knowledge, this is the first study on the vector competence of *Ae. koreicus* for flaviviruses. The transmission of ZIKV was similar to CHIKV in that it was also temperature-dependent. Transmission was only observed at 27 ± 5 °C but not at 24 ± 5 °C. Interestingly, the IR at 27 ± 5 °C was only slightly higher at 81.4% than 78.9% at 24 ± 5 °C, while the mean titer of the bodies was higher at 27 ± 5 °C with 10^5.43^ RNA copies/specimen compared to 10^4.47^ RNA copies/specimen at 24 ± 5 °C. These results are in line with results for the vector competence of the closely related species *Ae. japonicus*, which forms a monophyletic taxon with *Ae. koreicus* [44]. *Aedes japonicus* was also able to transmit ZIKV, and transmission is, likewise, temperature-dependent [45,46,47].

No transmission of WNV could be observed, neither at 27 ± 5 °C nor at 24 ± 5 °C. Interestingly, the IR was quite high at both temperatures with 85.4% (27 ± 5 °C) and 90.6% (24 ± 5 °C), as well as the mean body titer, with 10^6^ RNA copies/specimen for both temperatures.

Presumably, WNV was able to cross the midgut barrier after a blood meal and infect the whole mosquito body but was not able to cross the salivary gland barrier. Either the infection of the salivary glands failed or the escape of the virus from the tissue into the saliva failed. Both opportunities are described by Sanchez-Vargas et al. for different arboviruses in *Ae. aegypti* mosquitoes [48]. Even if body titers of arboviruses are high, such as the WNV titers we measured in our study, transmission is not consequential. Studies on the relationship between infection of different tissues and the presence of WNV in the saliva of *Culex quinquefasciatus* mosquitoes also did not show a clear correlation of high titers in the tissue and positive saliva. It was shown that the thorax titer of mosquitoes with positive saliva were significantly higher compared to WNV-negative saliva, but only 14 dpi, 21 dpi no difference of the thorax titers could be detected [49]. Although transmission of WNV is often related to mosquitoes of the *Culex* taxa, laboratory experiments with the closely related species *Ae. japonicus* revealed vector competence for WNV [50]. Albeit, there is also a study of Huber et al., where they investigated *Ae. japonicus* from Germany and did not detect any infection with WNV [51].

The interaction between a virus infection and the antiviral immune response of mosquitoes is highly specific and has to be evaluated for every species-virus combination. Even different virus strains and mosquito populations can have a high impact on vector competence [52]. To consider some natural factors, which can have an influence in vector competence studies, it is advantageous to use field-caught mosquitoes because these specimens harbor a natural microbiome. Several metagenomic studies of the microbiome and virome of mosquitoes yielded an important influence on whose composition for vector competence. Recent insights into the virome of mosquitoes revealed an ISV infection in all investigated species so far. Likewise, we were able to detect a novel ISV by NGS analysis in *Ae. koreicus*: Wiesbaden virus (WBDV) (Figure 1). Previous metagenomic studies of *Ae. koreicus* have focused only on the microbiome characterization [53].

The Luteo–Sobemo group comprises of *Luteoviridae* and *Sobemoviridae* virus families, infecting plants, arthropods, nematodes, molluscs and protists, with the majority of viruses in this group being formally unassigned [54]. We identified a novel member of the here described mosquito-specific Luteo–Sobemo group within the phylogeny of this unclassified virus group. The genome of WBDV contained two segments, encoding the replicase and the capsid. The replicase segment contained a ribosomal frameshift site before the coding regions of RdRp, which is typical for members of the Luteo-Sobemo-like group [54]. The phylogenetic position of WBDV in relation to those previously described in relatives and the high prevalence observed suggest that this virus most likely directly infects *Aedes* mosquitoes.

*Aedes koreicus* eggs were collected in the field, and mosquitoes were grown in the lab. Analyses of adults reared together as larvae’s showed WBDV-positive and WBDV-negative mosquitoes, hence transmission between larvae or adults seemed unlikely. Therefore, the assumption of ovarian transmission of WBDV virus is obvious, particularly with regard to other studies, where transovarian transmission for ISVs is also described [55]. Whether this is the only route of transmission remains unclear. Screening of ten different mosquito species for WBDV identified infected specimens for *Ae. japonicus* and *Ae. geniculatus*, suggesting that WBDV is most likely associated with container-breeding *Aedes* mosquitoes. Remarkably, all three WBDV-positive species are container breeders regularly found in the same habitat [56]; in fact, the *Ae. koreicus* and *Ae. geniculatus* specimens screened here were collected at the same place in 2020. *Aedes koreicus* and *Ae. japonicus* were investigated in 2018 and 2020 and showed positive results in both years. This leads us to hypothesize that this ISV is likely more habitat-specific than species-specific. Further analysis of the habitat, e.g., different insects, surfaces, microorganisms, etc., should be done to underline this hypothesis. Additionally, it would be interesting to test whether other mosquito species could be experimentally infected with WBDV.

The first studies on the coinfection of different mosquito-associated viruses and arboviruses revealed controversial results on the influence on the vector competence of coinfected mosquitoes. While some studies showed a suppression of WNV infection if mosquitoes are previously infected with mosquito-associated viruses, there was no influence observed in other studies [28,29]. We screened all arbovirus-infected *Ae. koreicus* specimens for coinfection with WBDV. Coinfection with WBDV and CHIKV showed a significantly positive correlation for infection, as well as for a higher CHIKV-titer of coinfected mosquitoes. Coinfection with ZIKV did not result in significant differences. The detection of novel ISVs is often linked to the chance of a potential role of the new ISV as a biocontrol agent by reducing the potential of arbovirus transmission, similar to the strategy for *Wolbachia* [26]. For example, coinfection with the ISV Cell-Fusing Agent Virus and dengue virus or ZIKV resulted in a reduction of arbovirus transmission [57,58]. The results of our study suggest another assumption: the invasion of a new habitat by a mosquito species can also enhance the vector potential of that species, for instance, by an infection with a new ISV, which might be true in our study. Due to the lack of data about the virome of other *Ae. koreicus* populations, we cannot state if only the *Ae. koreicus* from Germany are partly infected by WBDV and therefore have an enhanced CHIKV transmission potential, or if other *Ae. koreicus* populations are also infected by WBDV. Our study demonstrated that an ISV infection of mosquitoes can influence the IR and the virus titer in the mosquito body; hereby, an ISV infection can influence the vector competence of *Ae. koreicus.* Therefore, virome characterization of field-caught mosquitoes in vector competence studies should not be disregarded. Whether other mosquito species also show an increased vector potential for CHIKV upon coinfection with WBDV still needs to be addressed. Furthermore, it is of special interest whether the study results can be generalized, i.e., if WBDV infection enhances alphavirus infection and has no influence on flavivirus infection.

## 5. Conclusions

Surveillance of mosquitoes, especially of new invasive species such as *Ae. koreicus*, should be performed regularly to estimate the risk of local arbovirus transmission. In addition, the vector competence of a certain species for an arbovirus should be evaluated. Our study showed that the new European invader *Ae. koreicus* has the potential to transmit arboviruses, such as CHIKV and ZIKV, but transmission depends on high temperatures. We could not observe a transmission of WNV. In addition, we identified a new ISV named WBDV, which likely has a boost effect on CHIKV infection but not on ZIKV infection.

## Figures and Tables

**Figure 1 viruses-13-02507-f001:**
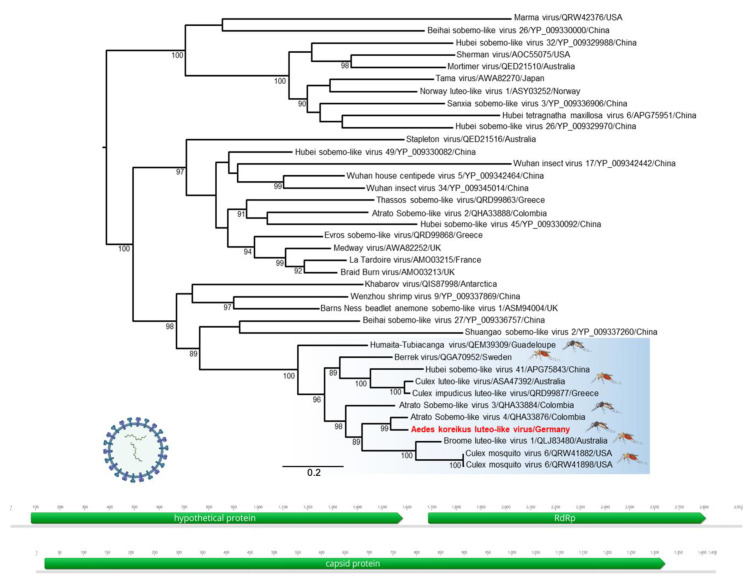
Phylogenetic analysis of luteo-/sobemo-like viruses. The maximum likelihood tree was constructed using RdRp protein sequences. Wiesbaden virus is highlighted in red. Blue highlights: This group of viruses was only detected in mosquitoes; all other clades are identified in insects or plants. Pictogram labeled with blood-sucking mosquitoes: genus *Culex*, non-blood-sucking mosquitoes: genus *Aedes*. Inset: Genome organization of Wiesbaden virus. Open reading frames (ORF) are indicated by green arrows, while nucleotide positions are shown above the genome. RdRp: RNA-dependent RNA polymerase. The scale bar represents amino acid substitutions per site, and bootstrap support values are displayed at the nodes (≥70%).

**Figure 2 viruses-13-02507-f002:**
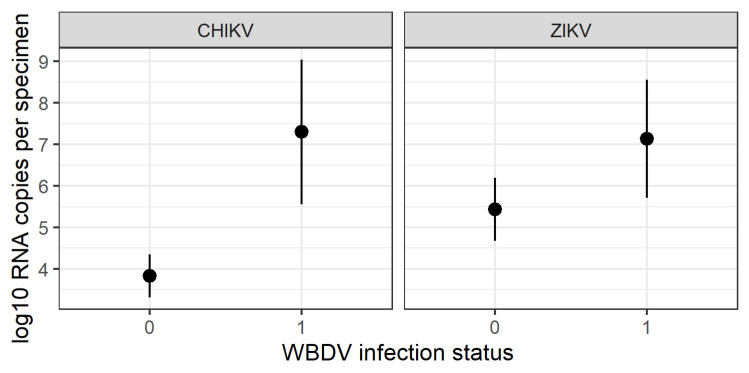
Body titers (log 10 RNA copies) of chikungunya Virus (CHIKV) and Zika Virus (ZIKV) (bodies of 24 ± 5 °C and 27 ± 5 °C were combined; 14 days post-infection) for *Aedes koreicus* specimens coinfected (1) and non-coinfected (0) with Wiesbaden virus.

**Table 1 viruses-13-02507-t001:** Infection rates (IR), transmission rates (TR), transmission efficiency (TE) and body titer of *Aedes koreicus* at two different temperatures (14 days post-infection) for arbovirus infection with chikungunya Virus (CHIKV), West Nile Virus (WNV) or Zika Virus (ZIKV), and information about IR with Wiesbaden virus (WBDV) and coinfection rate (CR).

Arbovirus	Temperature °C	IR Arbovirus	TR Arbovirus	TE Arbovirus	Mean (95% Confidence Interval) log10 RNA Copies/Specimen	IR WBDV	CR
CHIKV	24 ± 5	17.6% (6/34)	0% (0/6)	0% (0/34)	1.37 (0.29–2.45)	8.8% (3/34)	8.8% (3/34)
CHIKV	27 ± 5	68.2% (103/151)	6.8% (7/103)	4.6% (7/151)	4.68 (4.15–5.22)	6.6% (10/151)	6.0% (9/151)
WNV	24 ± 5	90.6% (29/32)	0% (0/29)	0% (0/32)	7.09 (6.15–8.03)	3.1% (1/32)	3.1% (1/32)
WNV	27 ± 5	85.4% (35/41)	0% (0/35)	0% (0/41)	6.7 (5.72–7.68)	0% (0/41)	0% (0/41)
ZIKV	24 ± 5	78.9% (30/38)	0% (0/30)	0% (0/38)	4.97 (4.02–5.92)	0% (0/38)	0% (0/38)
ZIKV	27 ± 5	81.4% (35/43)	5.7% (2/35)	4.7% (2/43)	6.2 (5.19–7.21)	20.9% (9/43)	20.9% (9/43)

## Data Availability

The data presented in this study are available on request from the corresponding author.

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
