# Peer review of "Vector Competence of the Invasive Mosquito Species Aedes koreicus for Arboviruses and Interference with a Novel Insect Specific Virus"

_viruses, 2021, doi:10.3390/v13122507_

Round 1

Reviewer 1 Report

Overall, I was able to read a manuscript made of very valuable and exciting work. There is no doubt how relevant and important the investigated topic is nowadays. In the light of globalization and climate change, the introduction of exotic mosquito species to new territories by humans (anthropogenic activities, trade, and travel) and the public health risk they pose by their pathogen vector capacity becomes more and more in focus. Although several Aedes invasive mosquitoes are known introduced to Europe, most of the investigations focus on the Asian tiger mosquito (Aedes albopictus) and the bush mosquito (Aedes japonicus) only. Notable, Aedes koreicus also seems to spread very fast across Europe, and adapt better to urban habitats than other invaders, our knowledge on this species’ ecology, including its vector capability for pathogen transmission in its new environment is still poor.

In the present study, the authors investigated the vector competence of Ae. koreicus for the most relevant human pathogenic mosquito-borne viruses we are (or will be soon) faced within Europe (CHIKV, ZIKV, WNV). Although laboratory conditions are difficult to reflect different environmental conditions and natural processes, such kind of vector competence studies provide extensive information and contribute to predicting potential public health risks in the case of novel exotic mosquitoes. Furthermore, the authors analyzed the virome of the investigated Ae. koreicus mosquitoes, thus provide viral metagenomic and metatranscriptomic data on that species for the first time.

The methods applied in this study are appropriate, including classical techniques, indicative factors, state-of-the-art sequencing techniques, and statistical models as well. The description of this section is detailed enough.

The results are properly articulated, following the logical sequence constructed in previous sections of the paper. Data are visualized very well with graphs and plots, helping understand the main findings and relations between the investigated parameters. When evaluating the results, the authors make logical, well-founded, and sufficiently critical conclusions, from which it is obvious that they have a good understanding of the significance and limitations of each result.

I have only a few specific questions:

  • experiments were performed under different climate conditions. Why did you choose the given temperatures (27°±5°C or 24°±5°C) and humidity (70%)? What do you think, different climatic conditions in Europe (e.g. in Italy or the Black Sea coast) may alter your findings, i.e. vector capability of different mosquito populations?
  • The Ae. koreicus population in Germany was described as the morphological variant of the species originating from mainland Korea, while all other populations in Europe show the morphological characteristics of the Jeju-do Island population (South Korea). It is conceivable that different morphological variants behave differently and have slight differences in their ecology (like in the case of Culex pipiens biotypes pipiens and molestus)? If yes, could it affect its vector competence for arboviruses?
  • Considering the different morphological variants, the impact of the novel Wiesbaden virus on arbovirus transmission can be generalizable for other European koreicus populations? It would be nice to discuss these circumstances in the manuscript.

Author Response

I have only a few specific questions:

  • experiments were performed under different climate conditions. Why did you choose the given temperatures (27°±5°C or 24°±5°C) and humidity (70%)? What do you think, different climatic conditions in Europe (e.g. in Italy or the Black Sea coast) may alter your findings, i.e. vector capability of different mosquito populations?

We have chosen 27°±5°C as start temperature of our studies, because it is known that ZIKV and WNV are temperature-depend transmitted, highest transmission occurs on tropical temperatures like 27°±5°C. After detecting transmission at 27°±5°C, we reduced the temperature to 24°±5°C, to test whether transmission occurs also at a lower temperature. Because no transmission was observed, studies were terminated at this temperature.

We have chosen 70% humidity because this is the maximum our incubators in the BSL3-insectary are able to provide and humidity is important for the survival of mosquitoes.

  • The Ae. koreicus population in Germany was described as the morphological variant of the species originating from mainland Korea, while all other populations in Europe show the morphological characteristics of the Jeju-do Island population (South Korea). It is conceivable that different morphological variants behave differently and have slight differences in their ecology (like in the case of Culex pipiens biotypes pipiens and molestus)? If yes, could it affect its vector competence for arboviruses?

Yes, this it is conceivable that different morphological variants have differences in their ecology like the mentioned Culex mosquitoes. Albeit in our knowledge, the only differences so far described for Ae. koreicus are morphological, no genomic differences like for the Culex are known neither any ecological differences are known. Beside this lacking information, the vector competence could be affected, like for example the vector competence of Culex pipiens biotypes pipiens and Culex molestus for WNV differs in the TR value, even both are able to transmit WNV (Jansen et al. 2019).

  • Considering the different morphological variants, the impact of the novel Wiesbaden virus on arbovirus transmission can be generalizable for other European koreicus populations? It would be nice to discuss these circumstances in the manuscript.

We don´t know if other Ae. koreicus populations are WBDV-positive, but it would be interesting to find out, especially with regard to the different morphological variants. Likewise, it would be very interesting to see, if the effect of WBDV infection which we observed would also take place in other Ae. koreicus populations or even for other mosquito species like Ae. geniculatus. So far, we are not able to infect mosquitoes in the lab with WBDV, therefore we are not able to conduct this kind of experiments.

Reviewer 2 Report

l127: what are the groups which were realised?

l138: concerning the temperature. The authors specify major difference concerning results in infection rates depending the temperature when variation of 3 °C are observed. The precision of temperatures is + or - 5°C. This is higher than difference between the 2 observed temperatures. Does this variation due to incubator precision? Real variations? Does a mean temperature measure realise?

With this kind of variations, it is not possible to conclude on the impact of temperatures on infection.

L 180: first use of the acronym WBDV without description

L 183: what is the material and methods concerning cell culture (how many passages, evaluation of positivity, increase of Cq between 2 passages…)

L221: TR, TE and IR first use of the acronyms WBDV without description

Figure 1: pooor quality. It is hard to distinguish the name.

L257: Could the author rephrase the sentence:” The proportion of the CHIKV-positive specimens with 92.3% (12/13) vs. 56.1% 257 (97/172) were statistically significant higher for WBDV-positive specimens (Chi-square 258 test: χ2 = 5.1121, df = 1, P = 0.02), while no significant effect was found for ZIKV (100% 259 (9/9) vs. 77.8% (56/72); Chi-square test: χ2 = 1.2875, df = 1, P = 0. 26) (table 1). Titres for 260 CHIKV were statistically significant higher in WBDV-positive specimens (Deviance = 261 145.81, df = 1, P = 0.004), but no correlation was observed for ZIKV (Deviance = 23.156, df 262 = 1, P = 0.12) (Fig.2)”. The compared parameters are unclear CHIKV positive specimens

L257 : What is the significance on a population of 13 individuals?

It is hard to understand the origin of the 12/13, 97/172… Could the authors explicit it?

L287: Could the authors reformulate the sentence “As in our study, transmission efficiency was also in the 287 lower-level of around 5% (3 and 10 dpi)”

L305: Could the authors reformulate the sentences from lines 305 to 310. They are quite complicated to understand

Author Response

l127: what are the groups which were realised?

Rephrased for better comprehension

l138: concerning the temperature. The authors specify major difference concerning results in infection rates depending the temperature when variation of 3 °C are observed. The precision of temperatures is + or - 5°C. This is higher than difference between the 2 observed temperatures. Does this variation due to incubator precision? Real variations? Does a mean temperature measure realise?

Explained in more detail. The 24h fluctuating temperatures of +/-5°C mimic the variation between day and night. 27°+/-5°C i> the mean temperature of a 24h cycle is 27°C and fluctuates between 32°-22°C and 24°+/-5°C is a 24h cycle between 29°-19°C with a mean temperature of 24°C.

With this kind of variations, it is not possible to conclude on the impact of temperatures on infection.

Changed accordingly, see comment L138

L 180: first use of the acronym WBDV without description

Changed accordingly

L 183: what is the material and methods concerning cell culture (how many passages, evaluation of positivity, increase of Cq between 2 passages…)

Information added

L221: TR, TE and IR first use of the acronyms WBDV without description

Changed accordingly

Figure 1: pooor quality. It is hard to distinguish the name.

A new version is uploaded

L257: Could the author rephrase the sentence:” The proportion of the CHIKV-positive specimens with 92.3% (12/13) vs. 56.1% 257 (97/172) were statistically significant higher for WBDV-positive specimens (Chi-square 258 test: χ2 = 5.1121, df = 1, P = 0.02), while no significant effect was found for ZIKV (100% 259 (9/9) vs. 77.8% (56/72); Chi-square test: χ2 = 1.2875, df = 1, P = 0. 26) (table 1). Titres for 260 CHIKV were statistically significant higher in WBDV-positive specimens (Deviance = 261 145.81, df = 1, P = 0.004), but no correlation was observed for ZIKV (Deviance = 23.156, df 262 = 1, P = 0.12) (Fig.2)”. The compared parameters are unclear CHIKV positive specimens

rephrased

L257 : What is the significance on a population of 13 individuals?

Changed accordingly, numbers are explained

It is hard to understand the origin of the 12/13, 97/172… Could the authors explicit it?

Changed accordingly (see comment L257)

L287: Could the authors reformulate the sentence “As in our study, transmission efficiency was also in the 287 lower-level of around 5% (3 and 10 dpi)”

Changed accordingly

L305: Could the authors reformulate the sentences from lines 305 to 310. They are quite complicated to understand

rephrased